# Body Composition, Anemia, and Kidney Function among Guatemalan Sugarcane Workers

**DOI:** 10.3390/nu13113928

**Published:** 2021-11-02

**Authors:** Lyndsay Krisher, Jaime Butler-Dawson, Karen Schlosser, Claudia Asensio, Elisa Sinibaldi, Hillary Yoder, Lynn Dexter, Miranda Dally, Daniel Pilloni, Alex Cruz, Diana Jaramillo, Lee S. Newman

**Affiliations:** 1Center for Health, Work & Environment, Department of Environmental and Occupational Health, Colorado School of Public Health, University of Colorado, Anschutz Medical Campus, Aurora, CO 80045, USA; JAIME.BUTLER-DAWSON@cuanschutz.edu (J.B.-D.); LYNN.DEXTER@CUANSCHUTZ.EDU (L.D.); miranda.dally@cuanschutz.edu (M.D.); DIANA.JARAMILLO@CUANSCHUTZ.EDU (D.J.); lee.newman@cuanschutz.edu (L.S.N.); 2School of Nutrition, Universidad Francisco Marroquin, 01010 Guatemala City, Guatemala; karensm@ufm.edu (K.S.); emsinibaldi@ufm.edu (E.S.); 3Pantaleon Group, 01010 Guatemala City, Guatemala; Claudia.asensio@gmail.com (C.A.); williams.pilloni@pantaleon.com (D.P.); acruz@pantaleon.com (A.C.); 4Division of Kinesiology & Health, University of Wyoming, Laramie, WY 82071, USA; hayoder@crimson.ua.edu; 5Division of Pulmonary Sciences and Critical Care Medicine, School of Medicine, University of Colorado, Anschutz Medical Campus, Aurora, CO 80045, USA

**Keywords:** kidney disease, nutrition, agricultural workers

## Abstract

Rates of anemia among agricultural workers, who are also at risk for kidney injury and chronic kidney disease of unknown cause (CKDu), are unknown. We evaluated body composition through the sum of three skinfolds among 203 male sugarcane cutters and assessed the relationship of variables related to nutrition, anemia (hemoglobin < 13 g/dL), and elevated hemoglobin A1c (HbA1c ≥ 5.7%) with estimated glomerular filtration rate (eGFR) using linear regression. Eleven percent of workers were at the level of essential body fat (2–5%). Anemia was present among 13% of workers, 70% of which were normochromic normocytic, a type of anemia suggesting potential underlying chronic disease. Anemia was more common among those with lower BMI and fat free mass. The prevalence of elevated HbA1c was 21%. A moderate negative correlation was found between hemoglobin and HbA1c (Pearson’s r = −0.32, *p* < 0.01) which suggests that HbA1c values should be interpreted with caution in populations that have high rates of anemia. Twelve percent of workers had reduced kidney function with an eGFR < 90 mL/min/1.73 m^2^. On average, the eGFR was 18 mL/min per 1.73 m^2^ lower [(95% CI:−24, −12), *p* < 0.01)] for those with anemia than those without, and 8 mL/min per 1.73 m^2^ lower among those with elevated HbA1c [(95% CI: −13, −2), *p* < 0.01]. Results will inform future studies examining the role of anemia in the evaluation of CKDu and interventions to improve nutrition for workers in low-resource settings.

## 1. Introduction

Guatemala faces challenges in regard to nutrition, including around food insecurity, lack of healthy food access and anemia that have resulted in the highest rates of chronic malnutrition, or stunting (low height-for-age) in Latin America [1,2,3]. While there is ample evidence of malnutrition and its impact on women and children in Latin America—for example, anemia affects over 30% of children under five, roughly a quarter of pregnant women, and 14% of women of reproductive age [1]—much less is known about the nutritional consequences for working age men in the region. 

Anemia is a known comorbidity of chronic kidney disease (CKD) as well as a risk factor for acute kidney injury (AKI), inadequate renal recovery, CKD progression, and associated mortality [4,5,6,7,8]. CKD of all causes is rising across the world, including in Guatemala [9]. Guatemala has been cited as having high rates of CKD and renal failure mortality, at 14.7 per 100,000 in 2013, of which 39% occurred in individuals less than 60 years of age [10]. Investigations are underway to address an epidemic of kidney injury and chronic kidney disease of unknown origin (CKDu) that affects agricultural workers, in particular sugarcane workers, who labor under conditions of high heat and humidity, including in Guatemala [11,12,13,14,15]. Currently, most research has focused on the etiologic roles of heat stress and environmental and occupational nephrotoxins, although the exact causes of the epidemic remain undetermined. While some published evidence in the CKDu literature suggests a contributory role for workers’ nutrition, namely anemia, in CKDu pathogenesis, it has not been frequently examined in epidemiologic studies. A study among 119 non-dialysis CKDu patients in Sri Lanka found a 72% prevalence of anemia, with a high proportion of those related to iron deficiency [16]. Similarly, in a study of Nicaraguan agricultural workers, anemia was a major risk factor for progression from acute renal injury to CKDu [8,17]. Anemia was also a common finding among CKDu patients in a study among workers from other industrial sectors in Nicaragua [18] and among sugarcane workers in El Salvador [19]. Because anemia may compound the effects of hypoperfusion and oxygen delivery to the kidney, anemia has been hypothesized to contribute to the development of CKDu in the context of heat stress and chronic inflammation [20]. Anemia has not been investigated as a risk factor for kidney dysfunction among workers in Guatemala, a region of high CKDu endemicity. 

Iron deficiency is the most common cause of anemia worldwide, typically driven by inadequate consumption of bioavailable iron [21,22,23], followed by other vitamin and mineral deficiencies including folate, vitamins B12 and A, acute and chronic inflammation, parasitic infections, and genetic or acquired disorders [24]. Rates of anemia, undernutrition and malnutrition among working males in Guatemala are unknown, including agricultural workers, since much of this work is in the informal sector and these workers do not receive regular health screening [25]. In a previous investigation, Guatemalan sugarcane field workers were found to experience negative anthropometric and weight changes across the six-month harvest, indicating workers’ diets may be hypocaloric relative to their energy expenditures [26]. In that assessment, researchers found workers to be very lean even at the start of the harvest season, with some at the level of essential body fat (2–5%) [27] and observed that workers lost weight, fat and fat free mass (an estimate of muscle mass) during the season despite being provided meals and snacks by their employer. Local workers from the coastal communities around the sugarcane mill in Guatemala were at higher risk of such losses, compared to seasonal workers who migrated from highland communities and lived onsite at the mill during the harvest. The loss of essential body fat can compromise kidney and other physiological function [26,27] with studies suggesting that the local workers also appear to be at higher risk for kidney dysfunction [11,13]. In addition, previous research in Guatemalan sugarcane workers has identified high rates of hypertension and of prediabetes, as measured by hemoglobin A1c (HbA1c) [13,15]. While this evidence suggests elevated rates of prediabetes in these populations, consideration has not been given to the possibility that elevated HbA1c levels are the result of laboratory false positives due to anemia [28,29]. Since these conditions are known risk factors for CKD, we sought to explore this association further in this study. 

The main objective of this study was two-fold. First, we sought to assess hematological and anthropometric values indicative of poor nutrition, including anemia, and underlying health status among a cohort of 203 sugarcane workers originating from both coastal and highland communities. Second, we sought to assess the relationship of hematological and anthropometric measures with kidney function. We hypothesized that there would be an association between these indicators and kidney dysfunction. We further hypothesized that anemia may partly explain the high prevalence of elevated HbA1c in this worker cohort. The overall goal of this study was to investigate the current nutritional status of the workers, including potential dietary inadequacies, in order to inform future workplace nutrition programs.

## 2. Materials and Methods

Data collection for this cross-sectional study was carried out by Pantaleon, a Guatemalan sugarcane agribusiness, and a team from Universidad Francisco Marroquín (UFM) as a quality improvement project during two workdays in January 2019, midway through the sugarcane harvest season. Sugarcane cutters were recruited from four randomly selected 50-worker work groups (out of a total of 91 groups) employed to cut sugarcane during the 2018–2019 sugarcane harvest by Pantaleon. A total of 203 male field workers participated. 

The sugarcane harvest lasts from November through May each year. Workers from two of the work groups were from local, coastal communities surrounding the sugarcane plantation (referred to as Zona workers). Workers from the other two groups were originally from highland region communities and were living onsite at the sugarcane mill for the duration of the harvest (referred to as Altiplano workers). Zona workers eat breakfast and dinner at home and bring their own food from home to work each day. Altiplano workers live in dormitory housing and are provided with three meals per day at a cafeteria near the dormitory, or directly in the fields during the workday. An example menu, which was designed to provide Altiplano workers with approximately 4000 calories per day, is provided in the Appendix A. 

Work practices are previously described [11]. In brief, sugarcane cutters work a six-day workweek followed by one day off, working approximately eight hours per day with two hours of rest breaks including lunch. Sugarcane cutting in Latin America demands high levels of physical exertion in a hot, humid environment, and has been associated with a metabolic load of 261 W/m^2^ (6.8 kcal/min) [30]; caloric and hydration demands are therefore comparable to those of endurance athletes [31,32,33,34]. 

All study participants were at least 18 years of age, were male, were screened for employment in November 2018, started the season with an estimated glomerular filtration rate (eGFR) of ≥90 mL/min/1.73 m^2^, and had no major illnesses affecting their ability to work as sugarcane cutters. 

Anthropometric, clinical, and biochemical measurements were collected from the workers in the morning (pre-shift) and afternoon (post-shift). Pre-shift measures included blood pressure (at least three min of seated rest before the measurement, OMRON Healthcare, Inc. BP742N, Lake Forest Il), weight (Tanita UM028 3601, Tokyo, Japan) and height (Seca CE0123 stadiometer, Chino, CA, USA) with shirt and shoes removed, and body composition which was collected by trained UFM field staff taking sum of skinfold caliper measurements of the chest, triceps and subscapular (Creative Health Slim Guide, Plymouth, MI, USA). Body mass index (BMI), body density, fat mass, body fat %, and fat free mass (an estimate of muscle mass) were calculated [35]. A blood sample was collected from workers (at least three min of seated rest) at the end of the work shift by trained phlebotomists, transported on ice and analyzed the same day by the accredited Herrera Llerandi Laboratory in Guatemala City. Blood measures were analyzed as follows: serum creatinine and blood urea nitrogen (BUN) by automated standard techniques (Abbott, Architect CI4100), serum albumin by colorimetric (Roche, Cobas Integra 400), HbA1c by ionic exchange high pressure liquid chromatography (Bio-Rad, D-10), and a complete blood count (CBC) by automated standard techniques (Abbott, CELL-DYN Ruby). The final datasets were deidentified and provided to the Center for Health, Work & Environment at the Colorado School of Public Health (CHWE) for analysis. This study was reviewed and approved by the Colorado Multiple Institutional Review Board (COMIRB, #18–0957). 

Descriptive statistics were calculated for all measures and are presented using the mean and standard deviation (SD) or frequency and percent (%). The primary outcome was kidney function, estimated GFR (eGFR), calculated using the Chronic Kidney Disease Epidemiology Collaboration formula and using measures of serum creatinine [36]. Anemia was defined as blood hemoglobin < 13 g/dL [24]. Low hematocrit was defined as <42%. Low serum albumin was defined as <3.5 g/dL and elevated HbA1c as between 5.7 and 6.4%. Elevated blood pressure was defined as systolic ≥ 120 mm Hg and/or diastolic ≥80 mm Hg. We used results from the CBC to investigate the underlying cause of anemia. Normochromic normocytic anemia was defined by a mean corpuscular hemoglobin concentration (MCHC) and mean corpuscular volume (MCV) within normal ranges (defined as MCHC of 32–36 g/dL and MCV 82–92 fL). Macrocytic anemia was defined by a higher than normal MCV, whereas microcytic refers to lower than normal MCV; likewise, hypochromic refers to an MCHC that is lower than normal [37]. 

To address our first hypothesis, we used linear regression to evaluate the association between eGFR and our main variables of interest: anemia, low hematocrit, low serum albumin, BMI, body composition (body density, body fat %, fat free mass, and fat mass calculated using sum of three skinfolds [35]), elevated HbA1c and elevated blood pressure. T-tests were used to identify differences in individual clinical, demographic and biomarker data between Zona and Altiplano workers. For noncontinuous data, the chi-square or Fisher exact test was used for comparison. As a post-hoc analysis, we assessed these same variables as risk factors for a) anemia and b) elevated HbA1c using a logistic regression framework, presented as odds ratios (OR) and 95% confidence intervals (CI) while controlling for age and worker origin (Zona or Altiplano). To test our second hypothesis, we used Pearson’s correlation coefficients to examine the strength of relationship between hemoglobin and HbA1c. To account for multiple comparisons, we conducted a Bonferroni adjustment and set the alpha level at α = 0.01. All analyses were performed using SAS Release 9.4 (SAS Institute, Cary, NC, USA).

## 3. Results

### 3.1. Demographic and Anthropometric Measures

A total of 203 cane cutters participated in the study, including 103 Altiplano workers and 100 Zona workers. Table 1 displays the demographic and anthropometric data. The mean age was 28 years (SD: 7). Body composition measures including BMI, body density, body fat percentage, fat mass, and fat free mass were similar between Zona and Altiplano workers. Average percent body fat was 8.5% (SD: 4.0%). Based on BMI, 5% of workers were underweight (BMI ≤ 18.5 kg/m^2^), while 7% of workers were considered overweight (BMI ≥ 25 kg/m^2^). Based on body fat %, 11% were at the level of essential body fat (2–5%) [27]. 

### 3.2. Markers of Underlying Health and Anemia

Markers of underlying health are presented in Table 2. There was no observable difference in eGFR between Zona and Altiplano workers. The average eGFR was 114.2 mL/min/1.73 m^2^ (SD: 17.8). Overall, 12% of workers had reduced kidney function by mid-harvest when these data were collected (eGFR < 90 mL/min/1.73 m^2^). Serum BUN was significantly higher among Zona workers (14.6 mg/dL (SD: 3.5)) compared to Altiplano (12.3 mg/dL (SD: 3.5)). As shown in Figure 1, the HbA1c was higher among Zona workers compared to Altiplano (5.5% (SD: 0.3) versus 5.3% (SD: 0.3), respectively; *p*-value < 0.01). The prevalence of elevated HbA1c was 21% overall; significantly more Zona workers were in the elevated range compared to Altiplano (30% vs. 12%, *p* < 0.01). No participants were diabetic (HbA1c ≥ 6.5%). Both systolic and diastolic blood pressure were higher on average among Altiplano workers. Approximately half of the workers overall had elevated blood pressure (51%).

Table 3 presents the results of the CBC. As shown in Figure 1, mean hemoglobin trended lower among Zona workers (14.0 g/dL (SD: 1.3)) compared to Altiplano (14.3 g/dL (SD: 1.1)), *p* = 0.04. Thirteen percent of workers (n = 27) in the cohort were anemic. Anemia was more common among Zona workers (17%) compared to Altiplano (10%, *p* = 0.13). Low hematocrit was common among workers (N = 109; 54%). Erythrocyte (red cell) distribution width was significantly higher among Zona workers (12.9 (SD: 0.9)) compared to Altiplano (12.6 (SD: 0.8)), *p* < 0.01. Erythrocytes and platelets were significantly lower on average among Zona workers (*p* < 0.01). Measures of the MCV, MCH and MCHC were within normal ranges on average and were not significantly different between the groups. 

### 3.3. Risk Factors for Lower eGFR, Elevated HbA1c and Anemia

Table 4 displays results of analyses to identify factors associated with eGFR, controlling for age and worker origin (Zona vs. Altiplano). eGFR was found to be inversely related to albumin and BUN. Additionally, workers with anemia and workers with elevated HbA1c were found to have lower eGFR. On average, the eGFR was 7.7 mL/min per 1.73 m^2^ lower among those with elevated HbA1c than those without (95% CI: −13.1, −2.3, *p* < 0.01), 18.2 mL/min per 1.73 m^2^ lower for those with anemia (95% CI: −24.2, −12.2, *p* < 0.01) and 7.7 mL/min per 1.73 m^2^ lower for those with low hematocrit. For each 1-unit increase in albumin, eGFR was nearly 12 mL/min per 1.73 m^2^ lower (95% CI: −21.1, −2.7, *p* < 0.01). As a post-hoc analysis we stratified the origin and age adjusted model for elevated HbA1c by anemia. For workers without anemia, the coefficient for elevated HbA1c is −5 mL/min/1.73 m^2^ (95% CI: −10, −1, *p* = 0.03) compared to −3 mL/min/1.73 m^2^ (95% CI: −3, 21, *p* = 0.78) for workers with anemia.

Table 5 displays results of risk factor analysis for anemia, controlling for age and worker origin. The odds of anemia were just over three times higher on average among those with elevated HbA1c than in those without (95% CI: 1.35, 8.08, *p* < 0.01). Higher BUN (indicating potential kidney dysfunction) was associated with 1.3 times greater odds of anemia (95% CI: 1.13, 1.43, *p* < 0.01). Several measures were associated with reduced odds for anemia, including increasing eGFR, fat free mass, and BMI. For each 1 mL/min per 1.73 m^2^ increase (improvement) in eGFR, the odds of having anemia were approximately 5% less likely (95% CI: 0.92, 0.97, *p* < 0.01). For each 1 kg/m^2^ increase in BMI the odds of anemia were 31% lower (95% CI: 0.53, 0.89, *p* < 0.01). Similarly, for each 1 kg increase in fat free mass the odds of anemia were 12% lower (95% CI: 0.80, 0.96), *p* < 0.01). 

Table 6 presents the results of risk factor analysis for elevated HbA1c. For each 1 mL/min per 1.73 m^2^ increase in eGFR, the odds of having elevated HbA1c were approximately 3% less likely (95% CI: 0.95, 0.99, *p* = 0.01). The odds of having elevated HbA1c were just over three times higher on average among those with anemia than in those without (95% CI: 1.35, 8.05, *p* < 0.01). 

### 3.4. Correlation of HbA1c and Hemoglobin

To test our second hypothesis, we examined whether the higher levels of HbA1c among the participants may be explained in part by false positive laboratory results due to anemia. Displayed in Figure 2, as hemoglobin levels increased, HbA1c decreased. The analysis found a moderate negative correlation between hemoglobin and HbA1c (Pearson’s r = −0.32, *p* < 0.01).

### 3.5. Anemia Type and Kidney Function

Utilizing data from the CBC, we characterized the type of anemia experienced among the 27 workers with low hemoglobin (10 Altiplano and 17 Zona), displayed in Table 7. Overall, 70% of anemia was normochromic normocytic—the Altiplano workers mainly experienced normochromic normocytic anemia (90%) compared to 59% of Zona workers. Thirty- six percent of Zona workers’ anemia was either borderline macrocytic (very near the cut-off) or macrocytic, compared to one worker (10%) from the Altiplano. 

Based on the observed association between anemia and eGFR (Table 4), we conducted a post hoc analysis between eGFR and each type of anemia (Table 8). As described above, those with anemia had significantly lower eGFR compared to those non-anemic. The eGFR ranged from 32 to 135 mL/min/1.73 m^2^ for those with anemia compared to 67–137 mL/min/1.73 m^2^ for those without (*p* < 0.01). 

## 4. Discussion

This study presents some of the only published data on the relationships between indicators of nutrition and underlying health status with kidney function of male agricultural workers in Latin America, and is the first to examine Guatemalan sugarcane workers, a population at especially high risk for CKDu in the region. Workers in the study were very lean in terms of body composition and BMI, with a significant proportion at the level of essential body fat. Anemia was common among participants and was most common for those with lower kidney function, lower BMI, and lower fat free mass. Surprisingly, elevated HbA1c and elevated blood pressure were very common, despite an extremely lean workforce. In support of our first hypothesis, both anemia and elevated HbA1c were associated with reduced kidney function, although body composition measures and blood pressure were not. In support of our second hypothesis, the significant moderate negative correlation between hemoglobin and HbA1c suggests that the unexpectedly high rates of elevated HbA1c (commonly referred to clinically as ‘prediabetes’) may be partially explained by anemia in this population, since the presence of anemia can falsely elevate HbA1c [38,39,40]. However, the findings from the post-hoc analysis suggest that for a segment of the worker population, the effect of elevated HbA1c is not primarily due to reduced hemoglobin. This suggests that low hemoglobin is contributing to the observed elevation of HbA1c, but is not the only contributor to that elevation. We conclude that HbA1c values should be interpreted with caution in similar populations in which there is a high prevalence of anemia and to understand the limitations of HbA1c measurement [28,29]. This will help avoid misclassification of patients as having elevated diabetes risk and instead focus interventions on addressable causes of anemia. Finally, while Zona workers had significantly higher levels of HbA1c and BUN, we did not find significant differences in other markers of nutrition and kidney function between the two groups.

### 4.1. Nutritional Markers, Underlying Health and Kidney Function

Workers in this study were very lean, with 11% at the level of essential body fat. Essential body fat is present in the nerve tissues, bone marrow, and organs; therefore, loss of these fat stores can compromise physiological function, including that of the kidneys; it can cause shrinking of the organs, impair hormonal functioning, and lead to other serious illness [41,42]. In this study, anemia was more common among those with lower BMI and fat free mass. Indeed, while the cause is frequently multifactorial, a consistent risk factor for anemia includes low body weight [43]. Serum albumin is another commonly used indicator of nutrition status. Low serum albumin may also indicate kidney injury, along with inflammation and malnutrition [44]. In this case, serum albumin levels were within the normal range, on average. 

In this study we observed approximately 12% of sugarcane workers with reduced kidney function (<90 eGFR), measured three months into the harvest season, despite having started the season with an eGFR of at least 90 mL/min per 1.73 m^2^ at the time they were hired. This percentage is somewhat higher compared to the one other published study in this worker population that also presented prevalence of reduced kidney function at mid-season, which found 6% of workers with lower than 90 eGFR [14,45]. Of note, workers in the prior study included both Zona and Altiplano workers, and were evaluated at approximately the same period of the season, however were enrolled at the time as participants in a hydration, rest and shade intervention which included incentives to promote kidney health [14]. In addition, measurement of kidney function was conducted at different times of the day and using different methods in that study. In our current study, anemia and elevated HbA1c were both associated with lower kidney function, consistent with a previous study which found higher HbA1c to be a risk factor for cross-shift reduction in eGFR [15]. Overall, we found a high proportion of workers who would be classified clinically as prediabetic based on HbA1c, including 30% of Zona workers and 12% of Altiplano workers (21% overall). This is lower than a previous study in this population which found 50% of workers with elevated HbA1c [15]. By way of comparison, among a population of 377 Mexican American and Other Hispanic American males aged 18–40 reported by the U.S. National Health and Nutrition Examination Survey (NHANES) from 2017-March 2020, the mean value of HbA1c was 5.42% (SD: 0.80) and the prevalence of elevated HbA1c was 13.5% (compared to 21% in our cohort), while the average BMI was 29.37 kg/m^2^ (SD: 6.46) compared to 21.7 kg/m^2^ in our cohort [46]. One other study in the CKDu literature that has reported levels of HbA1c was recently published from Sri Lanka, which found nearly 60% of participants with elevated HbA1c, and another 16% in the diabetic range [47]. Raines and colleagues reported 3.7% of participants in the diabetic range among cases in a study in Nicaragua, however that paper did not report results for the prediabetic range, limiting our ability to draw comparisons [48]. We conclude that studies with direct measurement of diabetes status through HbA1c are needed in Mesoamerica. In our present study, anemia and lower eGFR were both independent risk factors for elevated HbA1c. As shown in Table 6, body fat % was not associated with elevated HbA1c, indicating that underlying metabolic disease is unlikely to be a main driver of HbA1c levels. As we found in our analysis, the presence of anemia only partially explains the finding, which has been demonstrated elsewhere in the literature. In several studies, iron deficiency anemia has shown to be associated with elevated HbA1c levels [8,38,39,40]. Another possible explanation relates to hyperuricemia, which we did not measure in this study but has been found to be associated with the development of prediabetes [49]. 

Approximately half of workers had elevated blood pressure, similar to another study in this population which found 47% of study subjects with mild hypertension (defined as systolic blood pressure ≥ 130 or diastolic blood pressure ≥80) [13]. This is concerning, since in that study, workers who entered the workforce for the first time with mild hypertension were more likely to experience declines in kidney function over time [13]. Zona workers were at higher risk for declines, suggesting that community and other non-occupational exposures (that may be further exacerbated by work) could play a role in the development of kidney dysfunction early in life [13]. In contrast, in the present study, which was cross-sectional as opposed to longitudinal, Altiplano workers had higher blood pressure on average. As with elevated HbA1c, elevated blood pressure was a surprising finding given our population of lean, physically active workers. As suggested by Dally et al., 2020, it may be that mild elevation in blood pressure is an early sign of CKD in this population. Furthermore, some evidence suggests that maternal undernutrition is associated with reduced renal capacity (possibly by reducing the size and number of nephrons) as well as increased blood pressure in offspring [50,51,52]. Further longitudinal research to examine the relationship of maternal nutrition with kidney function from childhood through working age is warranted. Of note, it is possible that some of the blood pressure results in our study could have been influenced by the setting (i.e., ‘white coat syndrome’) or of insufficient resting time prior to measurement. Additional measurements would be needed to clinically diagnose hypertension [53]. Even so, the combined findings of these studies point to the need for routine screening of blood pressure among worker and community populations in the region. 

### 4.2. Anemia

Overall, 13% of the cohort was anemic, based on low hemoglobin concentrations. The odds of having anemia were greater among workers with lower kidney function and among workers with elevated HbA1c, while higher BMI and fat free mass were associated with lower risk. More than half of workers had low hematocrit, which could signal nutrient deficiency anemia of vitamin B12, folate or iron. In terms of effects on kidney function, reduced hematocrit may impede delivery of oxygen to tubular cells [20]. Anemia is a frequent finding among CKDu patients in the literature, although the mechanism is undetermined [8,16,17,18,19]. Heat stress has been hypothesized as an important risk factor for the development of CKDu [14,19,20]. The negative effects of heat stress and chronic inflammation on the kidneys may be further compounded by anemia [20]. As indicated by Hansson and colleagues, measurement of erythropoietin levels may help to determine the role of anemia relative to tubular injury [20]. Of those in our cohort with hemoglobin-defined anemia, the majority were normochromic normocytic (MCV and MCHC in normal ranges), which is consistent with other studies [54]), with some differences by origin as described below. This type of anemia is frequently the result of an underlying chronic disease, including potentially renal insufficiency. While microcytic anemia is the more common type seen with iron deficiency, in clinical practice, normochromic normocytic anemia is sometimes observed in early stages of iron deficiency, as is a combination of both iron and B12/folic acid deficiency [55]. About one-third of the cases of anemia in Zona workers were either macrocytic or borderline macrocytic. Macrocytic anemia may be an indication of folate or vitamin B12 deficiency, or possibly an outcome of liver disease, alcohol toxicity, or myelodysplasia [37,56]. Finally, anemia is sometimes seen among heavy endurance athletes, including dilutional anemia from increased plasma volume, gastrointestinal bleeding leading to iron deficiency, intravascular hemolysis, or exercise-induced acute phase response with production of inflammatory cytokines [57,58,59]. 

### 4.3. Differences by Worker Origin

Among workers in this cohort, the only significant differences by origin were in HbA1c and BUN, of which Zona workers had higher levels. Significantly more Zona workers had elevated HbA1c (30%) compared to Altiplano (12%). Some of this difference may be the result of falsely elevated HbA1c due to anemia as described above; it could also be driven by underlying kidney dysfunction that has previously been documented among Zona workers [11,13], of which the higher levels of BUN may be a signal. Our prior assessment of anthropometric measures found that Zona workers were at higher risk for cross-season reductions in body weight, BMI and other body composition parameters [26]. Future work will confirm cross-season changes in this population and evaluate other community and individual risk factors that make Zona workers higher risk for such changes. As a pilot, we assessed nutritional intake using a food frequency questionnaire among a sample of workers and carried out direct observations of family food preparation at several Zona worker households. Worker food preferences were also collected, in preparation for future intervention. The results of the assessment, which are available in the Appendix A, found that Zona workers consume less protein and fewer calories than Altiplano workers, on average. While these findings will need to be confirmed among a larger cohort of workers, and compared with more precise data on energy expenditure, it is useful to inform community and workplace interventions aimed at improving nutritional status among workers and their families. 

### 4.4. Limitations and Future Directions

The cross-sectional design of this study is a limitation—we cannot assess whether the anemia seen in this population is indeed a risk factor for reduced kidney function, or instead an effect of existing kidney dysfunction among the workers. Future work will assess longitudinal trends in nutrition indicators and kidney function across the harvest. Nevertheless, this is one of the only studies to evaluate kidney function in sugarcane workers part way through the harvest season. We used hematological and anthropometric measures to infer nutrition status however micronutrient analyses would be needed to assess true deficiencies in iron (serum ferritin), B12 and folate. Diet for much of the Guatemalan population is based upon cereals (primarily from corn used to make tortillas), sugars, rice, and beans. These foods may meet most of the energy requirements for the general population but frequently lack protein and fats, macronutrients which are critically important for those performing heavy physical exertion [42], as well as micronutrients. Micronutrient deficiencies have been addressed at the national level to some degree through supplementation of staple foods, including the addition of iodine to salt, vitamin A to sugar, and the fortification of wheat flour with iron and b-complex vitamins (B1, B2, niacin, and folic acid) [1,60]. However, these nutrition and dietary practices may increase susceptibility to nephrotoxic contaminants that are present in the environment. Some staple foods can be contaminated by nephrotoxic heavy metals (e.g., rice by arsenic and cadmium) [61,62] or by aflatoxin in the case of corn [63]. Future work in this population should evaluate macro and micronutrient deficiency as well as evaluate potential exposures to nephrotoxic contaminants through the food supply.

While there are other potentially more precise methods to measure body composition available in laboratory settings, for this study skinfold caliper measurement was feasible in the field and was conducted by trained study personnel. Our conservative definition of anemia based on WHO guidelines may not have captured all true cases of anemia; some studies suggest that a higher cutoff of hemoglobin may be appropriate [64]. However, since blood was drawn at the end of the workday and thus increased plasma volume from overhydration may be a concern as described above, we chose to use the lower, more conservative definition. Similiarly, it is possible that hematocrit values were reduced somewhat by high levels of free water intake among the workers as we have seen in another study in this population [34,65], although we did not measure fluid intake, serum osmolality or cross shift weight change as part of this study to be able to confirm this effect. Efforts by Pantaleon to promote consumption of electrolytes (5 L per day) in lieu of free water intake (now recommended *ad libitum*) among workers in recent years have lessened our concern about this issue. We also did not measure cigarette smoking in this study. Future work will examine this and other individual and household risk factors for reduced kidney function that may be related to diet, nutrition, and cooking practices, such as through exposure to nephrotoxic heavy metals and agrochemicals ingested through contaminated food, water, and air. It will be important to understand how the occupational hazards of high energy expenditure work in hot climates that lead to CKDu might be compounded by these and other potentially modifiable risk factors.

## 5. Conclusions

Sugarcane workers carry out intense daily physical exertion and yet in Latin America, many are undernourished and anemic. Anemia was associated with lower eGFR and may explain high rates of elevated HbA1c found in this population in the absence of cases of diabetes. Future research should investigate these associations further. Anemia and undernutrition continue to pose a major burden on populations in developing countries like Guatemala [66]. The numerous health and economic consequences of anemia and dietary inadequacy among working people and families warrant the implementation and evaluation of practical workplace and community interventions to improve nutrition, productivity, and overall health.

## Figures and Tables

**Figure 1 nutrients-13-03928-f001:**
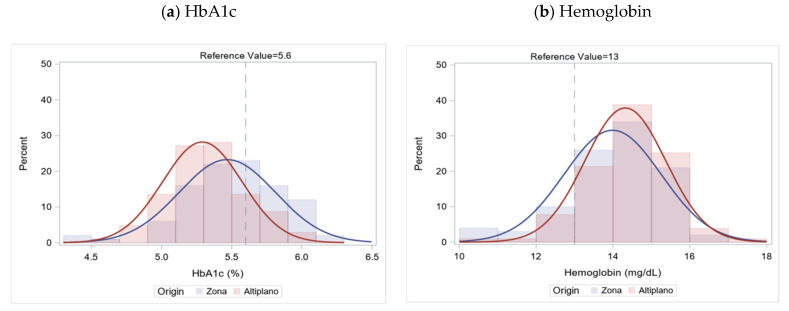
(**a**) Distribution of HbA1c and (**b**) Distribution of Hemoglobin, by worker origin, mid-harvest, January 2019.

**Figure 2 nutrients-13-03928-f002:**
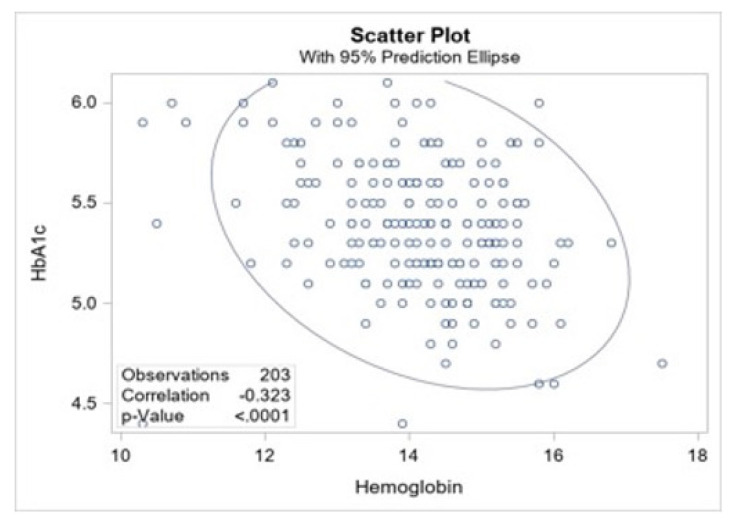
Scatter plot with 95% prediction ellipse depicting correlation between hemoglobin and HbA1c among 203 sugarcane workers.

**Table 1 nutrients-13-03928-t001:** Univariate comparisons of demographic and anthropometric measures between workers from highland communities (Altiplano) versus coastal (Zona), mid-harvest, January 2019.

Variable	All Participants (N = 203)	Altiplano (n = 103)	Zona (n = 100)	*p*-Value
	Mean (SD) or N (%)			
Age (years)	28 (7)	28 (7)	28 (8)	0.67
Height (cm)	160.2 (5.5)	159.5 (4.5)	160.8 (6.3)	0.09
Weight (kg)	55.7 (7.5)	55.5 (6.5)	55.9 (8.4)	0.72
Body mass index (kg/m^2^)	21.7 (2.5)	21.8 (2.1)	21.6 (2.7)	0.52
--Underweight (BMI ≤ 18.5), n(%)	10 (5%)	6 (6%)	4 (4%)	
--Overweight (BMI ≥ 25), n(%)	14 (7%)	7 (7%)	7 (7%)	0.90
Body density (kg/L)	1.1 (0.0)	1.1 (0.0)	1.1 (0.0)	0.99
Body fat %	8.5 (4.0)	8.5 (3.5)	8.5 (4.5)	0.96
--Essential body fat % (2–5%), n(%)	23 (11%)	9 (9%)	14 (14%)	0.23
Fat mass (kg)	5.0 (3.8)	4.9 (2.6)	5.0 (3.8)	0.71
Fat free mass * (kg)	50.8 (5.2)	50.7 (4.8)	50.9 (5.7)	0.78

* includes muscle, water, and bone mass.

**Table 2 nutrients-13-03928-t002:** Markers of underlying health among participants, mid-harvest, January 2019.

Biomarker	All Participants (N = 203)	Altiplano (N = 103)	Zona (N = 100)	*p*-Value	Reference Values
	Mean (SD) or N (%)				
Serum Creatinine	0.9 (0.2)	0.9 (0.2)	0.9 (0.2)	0.33	0.6–1.2 mg/dL
eGFR	114.2 (17.8)	113.2 (16.6)	115.2 (19.0)	0.44	≥90 mL/min/1.73 m^2^
--60–89 eGFR (n (%))	22 (11%)	11 (11%)	11 (11%)	0.93	60–89 mL/min/1.73 m^2^
--30–59 eGFR (n (%))	3 (1%)	1 (1%)	2 (2%)		30–59 mL/min/1.73 m^2^
Serum Albumin	4.8 (0.2)	4.8 (0.2)	4.8 (0.2)	0.44	3.4–5.4 g/dL
BUN	13.5 (3.7)	12.3 (3.5)	14.6 (3.5)	<0.01 *	7–20 mg/dL
--Elevated BUN (n (%))	12 (6%)	5 (5%)	7 (7%)	0.52	>20 mg/dL
HbA1c	5.4 (0.3)	5.3 (0.3)	5.5 (0.3)	<0.01 *	3.6–5.6%
--Elevated HbA1c (n (%))	42 (21%)	12 (12%)	30 (30%)	<0.01 *	5.7–6.4%
Systolic BP, mean (SD)	114.3 (12.3)	116.2 (11.6)	112.3 (12.8)	0.03	<120 mmHg
--High Systolic (n (%))	88 (44%)	52 (51%)	36 (37%)	0.05	≥120 mmHg
Diastolic BP, mean (SD)	70.6 (7.7)	71.5 (7.9)	69.6 (7.5)	0.09	<80 mmHg
--High diastolic (n (%))	39 (19.4%)	24 (23.3%)	15 (15.3%)	0.15	≥80 mmHg
Blood pressure (n (%))					
--Normal	98 (49%)	42 (41%)	56 (57%)	0.02	<120 and < 80 mm Hg
--Elevated	103 (51%)	61 (59%)	42 (42%)		≥120 and/or ≥ 80 mm Hg

* Values significant at *p* ≤ 0.01. *eGFR* estimated glomerular filtration rate (mL/min/1.73 m^2^), *BUN* blood urea nitrogen, *BP* blood pressure.

**Table 3 nutrients-13-03928-t003:** Complete blood count (CBC) measures.

Blood Count Measures	Cohort (n = 203)	Altiplano (n = 103)	Zona (n = 100)	*p*-Value	Reference Values
Mean (SD) or N (%)					
Hemoglobin	14.2 (1.2)	14.3 (1.1)	14.0 (1.3)	0.04	13–18 g/dL
--Anemia, n (%)	27 (13%)	10 (10%)	17 (17%)	0.13	<13 g/dL
Hematocrit	41.4 (3.3)	41.8 (3.0)	41.0 (3.6)	0.07	42–50%
--Low Hematocrit, n (%)	109 (54%)	51 (50%)	58 (53%)	0.23	<42%
MCV	90.4 (3.5)	90.1 (3.4)	90.8 (3.6)	0.11	82–92 fL
MCH	31.0 (1.3)	30.9 (1.3)	31.1 (1.3)	0.20	27–32 pg
MCHC	34.3 (0.5)	34.3 (0.5)	34.3 (0.6)	0.88	32–36 g/dL
RDW	12.8 (0.9)	12.6 (0.8)	12.9 (0.9)	<0.01 *	11.8–15.6%
% Lymphocytes	31.6 (7.1)	30.9 (7.6)	32.4 (6.6)	0.14	20–45%
# Lymphocytes	2.9 (0.7)	2.8 (0.6)	2.9 (0.7)	0.17	1.09–2.99 K/uL
% Monocytes	9.4 (3.9)	7.7 (1.3)	11.2 (4.7)	<0.01 *	3–8%
# Monocytes	0.9 (0.4)	0.7 (0.2)	1.0 (0.5)	<0.01 *	0.24–0.79 K/uL
% Segmented neutrophils	59.2 (8.5)	61.5 (8.3)	56.9 (8.1)	<0.01 *	40–60%
# Segmented neutrophils	5.6 (1.8)	5.9 (2.1)	5.21 (1.3)	<0.01 *	1.63–6.96 K/uL
Leukocytes	9.3 (2.0)	9.4 (2.2)	9.1 (1.7)	0.38	5–10 K/uL
Erythrocytes	4.6 (0.4)	4.7 (0.4)	4.5 (0.4)	0.01 *	4.5–6.2 M/uL
Platelets	324.7 (71.5)	339.9 (70.5)	309.0 (69.3)	<0.01 *	150–500 K/uL

* Values significant at *p* ≤ 0.01. *MCV* mean corpuscular volume, *MCH* mean corpuscular hemoglobin, *MCHC* mean corpuscular hemoglobin concentration, *RDW* erythrocyte distribution width. *%* percent, *#* number.

**Table 4 nutrients-13-03928-t004:** Association of hematological and anthropometric explanatory variables with eGFR among sugarcane cutters (n = 202), controlling for age and origin.

Measures	Estimate (95% CI)	*p*-Value
Elevated HbA1C (≥5.7) (ref: No)	−7.7 (−13.1, −2.3)	<0.01 *
Anemia (Hb < 13 g/dL)	−18.2 (−24.2, −12.2)	<0.01 *
Low hematocrit (<42%) (ref: No)	−7.7 (−12.0, −3.5)	<0.01 *
Serum albumin (g/dL)	−11.9 (−21.1, −2.7)	<0.01 *
BUN (mg/dL)	−2.9 (−3.3, −2.4)	<0.01 *
Systolic blood pressure (mmHg)	0.0 (−0.2, 0.2)	0.99
Diastolic blood pressure (mmHg)	−0.0 (−0.3, 0.3)	0.95
Body fat %	−0.1 (−0.6, 0.5)	0.79
Fat mass (kg)	−0.2 (−0.8, 0.5)	0.68
Fat free mass (kg)	−0.2 (−0.6, 0.2)	0.32
BMI (kg/m^2^)	−0.2 (−1.1, 0.7)	0.64

* Values significant at *p* ≤ 0.01. *BUN* blood urea nitrogen, *BMI* body mass index.

**Table 5 nutrients-13-03928-t005:** Risk factor analysis comparing participants with anemia (Hb < 13 g/dL, n = 27) versus participants with normal hemoglobin (n = 176), controlling for age and origin.

Measures	Odds Ratio (95% CI)	*p*-Value
eGFR (ml/min per 1.73 m^2^)	0.95 (0.92, 0.97)	<0.01 *
Elevated HbA1c (≥5.7%) (ref: No)	3.30 (1.35, 8.08)	<0.01 *
Serum albumin (g/dL)	0.12 (0.02, 0.78)	0.03
BUN (mg/dL)	1.27 (1.13, 1.43)	<0.01 *
Systolic blood pressure (mmHg)	0.98 (0.95, 1.02)	0.32
Diastolic blood pressure (mmHg)	0.98 (0.93, 1.04)	0.48
Body fat %	0.84 (0.72, 0.99)	0.03
Fat mass (kg)	0.75 (0.58, 0.96)	0.02
Fat free mass (kg)	0.88 (0.80, 0.96)	<0.01 *
BMI (kg/m^2^)	0.69 (0.53, 0.89)	<0.01 *

* Values significant at *p* ≤ 0.01. *eGFR* estimated glomerular filtration rate, *BUN* blood urea nitrogen, *BMI* body mass index.

**Table 6 nutrients-13-03928-t006:** Risk factor analysis comparing participants with elevated HbA1c (HbA1c ≥5.7, n = 42) versus participants with normal HbA1c (n = 161), controlling for age and origin.

Measures	Odds Ratio (95% CI)	*p*-Value
eGFR (mL/min/1.73 m^2^)	0.97 (0.95, 0.99)	0.01 *
Anemia (Hb < 13 g/dL)	3.30 (1.35, 8.05)	<0.01 *
Low hematocrit (<42%) (ref: No)	2.48 (1.16, 5.29)	0.02
Serum albumin	0.37 (0.08, 1.73)	0.21
BUN	1.14 (1.03, 1.26)	0.01 *
Systolic blood pressure (mmHg)	1.02 (0.99, 1.05)	0.24
Diastolic blood pressure (mmHg)	1.01 (0.96, 1.05)	0.85
Body fat %	1.02 (0.93, 1.11)	0.71
Fat mass (kg)	1.03 (0.93, 1.13)	0.63
Fat free mass (kg)	0.98 (0.92, 1.05)	0.62
BMI (kg/m^2^)	1.05 (0.92, 1.12)	0.49

* Values significant at *p* ≤ 0.01. *eGFR* estimated glomerular filtration rate, *BUN* blood urea nitrogen, *BMI* body mass index.

**Table 7 nutrients-13-03928-t007:** Anemia type among 27 sugarcane workers with low hemoglobin.

Anemia Type	Cohort (n = 27)	Altiplano (n = 10)	Zona (n = 17)
	N(%)		
Microcytic	0 (0%)	0 (0%)	0 (0%)
Hypochromic Microcytic	1 (4%)	0 (0%)	1 (6%)
Macrocytic	4 (15%)	1 (10%)	3 (18%)
Borderline Macrocytic	3 (11%)	0 (0%)	3 (18%)
Normochromic Normocytic	19 (70%)	9 (90%)	10 (59%)

**Table 8 nutrients-13-03928-t008:** Average eGFR by anemia type.

Anemia Type	eGFR, Mean (SD)	Range	*p*-Value
Non-anemic (Hb ≥ 13 mg/dL) (n = 175)	117 (14)	67–137	<0.01 *
Anemic (Hb < 13 mg/dL) (n = 27)	96 (28)	32–135	
--Hypochromic Microcytic (n = 1)	110 (-)	-	
--Macrocytic (n = 4)	89 (45)	32–135	
--Borderline Macrocytic (n = 3)	80 (19)	60–97	
--Normochromic Normocytic (n = 19)	100 (25)	36–134	

* Values significant at *p* ≤ 0.01. *eGFR* estimated glomerular filtration rate (mL/min/1.73 m^2^*)*, *Hb* Hemoglobin.

## Data Availability

The data presented in this study are available on request from the corresponding author.

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
