# Peer review of "Body Composition, Anemia, and Kidney Function among Guatemalan Sugarcane Workers"

_nutrients, 2021, doi:10.3390/nu13113928_

Round 1

Reviewer 1 Report

The study is on an important topic, the role of nutrition on kidney health among workers conducting physically demanding work in heat. The design has both strengths and limitations, a strength being one group with a more controlled, and seemingly adequate diet, and a comparison group eating what is typically eaten locally. Limitations include the cross-sectional design, and that there is little information on nutritional differences between the groups, making it likely that other causes than nutritional contribute to the differences seen here. It is valuable as a first analysis in its field, but there are several issues which needs to be resolved.

Method

“Prediabetes”. The question is whether HbA1c in this setting at all is driven by metabolic disease/insulin resistance, or whether Hb or continuous sugar liquid consumption are the main driver of elevated HbA1c levels. Referring to mildly elevated HbA1c as “prediabetes” in that case is misleading, as individuals with elevated HbA1c due to anemia are, at least as I am aware, not more likely to develop diabetes. What I would suggest is that the authors assess whether there at all is an association between % body fat and HbA1c in this population. An example of an analysis of this association is found here: https://academic.oup.com/jes/article/1/6/600/3738304. If that correlation between this key determinant of metabolic risk and insulin resistance and HbA1c is weak (I would suspect it is weaker than the correlation between Hb and HbA1c) that would serve as evidence that this population does not really have that big of a problem with metabolic disease/insulin resistance.

Rather than referring to mildly elevated HbA1c as “prediabetes” they should refer to it as “elevated HbA1c” throughout the manuscript, as the link between HbA1c and actual metabolic disease/insulin resistance and risk for the clinical disease entity diabetes in this population really is questionable. Especially so if the association between HbA1c and body fat cannot be established.

Also, “hypertension” is a clinical diagnosis reserved for those in which elevated blood pressure has been recorded at three occasions. Referring to those with elevated blood pressure at one occasion as hypertensive is inappropriate, they should be referred to as having had elevated blood pressure.

Page 4, line 161 “Paired” t-test. How could a paired t-test be used? There were no repeat measurements within the same worker when that same worker was both a Zona and Altiplano worker, or am I mistaken? The Zona and Alitplano workers were different workers, right? Or were they in any way matched, so that a conditioned, i.e. paired, test may have been appropriate? If so, this individual-level matching needs to have been explained before.

The sampling procedures need a better description. For hematocrit it needs to be clear what the body position was and for how long the rest was before sampling, as body position influence these parameters (https://link.springer.com/article/10.1186/s12968-018-0464-9). If this was not standardized it needs to be clarified.

Results

Figure 1. What does the bell-shape refer to? The combined distribution for both origins? Why not separate this bell-shape line by group? A bell-shape representing a corresponding sex, age and ethnic group in the US may be appropriate to introduce here, as that would place this population in a context familiar to many readers. The publicly available NHANES datasets, which include variables for age, sex, ethnicity/race, HbA1c, Hb and body fat % would be valuable to describe in which ways this population is different from a population in which there no apparent CKDu epidemic (US Hispanics males 18-~40 years old).

Table 2. I do not understand how the comparison at row 4 “30-59 eGFR” could give a p<0.01. There are 1/103 worker vs. 2/100 workers in the other group. My calculation of a chi-test gives a p-value of 0.62.

Table 4. How can a coefficient have both a 95% confidence interval which crosses 0 and a p-value <0.01? This page may be of interest: https://sphweb.bumc.bu.edu/otlt/mph-modules/ep/ep713_randomerror/ep713_randomerror6.html. Further, there seems to be something missing in the table title. Perhaps “Association between” first?

I am not sure I understand why origin was adjusted for - why does this confound the association between the parameters in the table and eGFR? It would be interesting to see also estimates not adjusting for origin. I believe the table is spacious enough for this, especially if removing some of the excess significant figures.

Table 5. Again, how can a coefficient have both a 95% confidence interval which crosses 1 have a p-value <0.01? https://sphweb.bumc.bu.edu/otlt/mph-modules/ep/ep713_randomerror/ep713_randomerror6.html.

Table 6. Heading. “N=42” is the number of workers with pre-diabetes, but it says in the end of the heading, so that it seems this is the total number of workers included in the table. 42 should be after prediabetes, and “N=161”(?) should be after “normal HbA1C”.

Table 8. There were n=81 non-anemic and n=27 anemic workers. This sums to 98. What happened to the remaining 105 workers included in the study?

Tables 2, 4, 8. eGFR is presented with 2 decimal points. These can be crossed out. I very much doubt that the creatinine measurements are accurate enough for this level of precision, and I am certain that the eGFR estimating equation is not so accurate that this level of precision in estimates of GFR can be achieved. It is just unnecessary ink and text for the reader. I encourage the authors to review the other parameters in the other tables to see if there are additional decimal points which are better left out too.

3.3. “Workers with prediabetes were found to have lower eGFR”. Aside from my questions on how a coefficient can have both a 95% C.I. and a confidence interval which crosses 0 and a p-value <0.01, I also wonder whether this association holds when adjusting for Hb levels or anemia. Before concluding that “prediabetes” contributes significantly to reduced eGFR in this setting, I would like the authors to make sure that “prediabetes” is not primarily due to reduced Hb (which likely is secondary to renal disease in itself).

Line 230: “related to HbA1C and BUN”. eGFR was related to HbA1c (although the confidence interval encompass 0 (see above)), but this is said in the next sentence as well (“Additionally… “workers with prediabetes”). I think it should say “albumin and BUN” on line 230.

3.5 Anemia type and kidney function

A scatter MCV and MCHC with color or symbol coding by place of residence may be easier to interpret than the present table. There could be lines indicating threshold values for macrocytosis etc. At present, the “Borderline Macrocytic” category is not well described, and it should be below “Macrocytic”.

Discussion

4.1 Lines 342-onwards, and also line 301 and onwards about associations between local residence, hypertension and eGFR. In research previously conducted by the researchers (ref 13, Dally et al 2020, Plos One), local residence (i.e. Zona workers) and mild hypertension were risk factors for eGFR decline among sugarcane workers in exact the same setting. In the present study, place of residence and hypertension are not at all associated with kidney function level at mid-harvest. This apparent discrepancy is not recognized in the present manuscript, although it needs to be acknowledged and possible reasons investigated.

I understand that the present manuscript only reports on a cross-sectional analysis while the previous study reported on a longitudinal analysis, and the study population in the previous study consisted only of newly hired. Nevertheless, considering the rather strong inferences the authors made on their previous finding of an association between hypertension, local residence and eGFR decline, i.e. that a pre-existing non-occupational cause predispose workers to CKDu, it is important that this is clarified in the available material. In the present material, did the workers with hypertension have a larger decline in eGFR from baseline to mid-harvest than those without? If not, what implications does this have for the authors’ previous conclusion that preexisting mild hypertension is a risk factor for CKDu? Is hypertension only a risk factor if measured at baseline in a previously non-employed population? Likewise, did local workers have a larger decline in eGFR from baseline to mid-harvest in this population, or is this longitudinal decline something which was only observed among the previously newly hired workers included in Dally et al 2020?

In the present manuscript, although it did not reach statistical significance at 0.01 (rather 0.05), hypertension was more common in Altiplano than Zona workers. The exact same analysis was not performed in the Dally 2020 paper, but it seems like the opposite would be the case in that population, as high blood pressure and Zona origin were both associated with being in the rapid eGFR decline group. It would be interesting to learn whether hypertension prevalence differs by origin also in the Dally 2020 material.

4.2. There needs to be a reference to the sentence that “While microcytic anemia is the more common type seen with iron deficiency, normochromic normocytic anemia can be observed in early stages”. I am not aware of this, but it may be. It needs to be referenced.

In clinical practice, it is sometimes considered that a mix of both iron and B12/folic acid deficiency may result in normocytic anemia. Are the authors aware of whether this has any basis in research? If so, is this possibility something to consider in the current setting where most anemia was normocytic?

4.3. The comparison between workers with a more or less controlled diet (Altiplano workers eating the mill breakfast, lunch and dinner) and those eating typical food in the local community is interesting and rightfully at the center for comparisons between anthropometric, health and nutritional indicators (i.e. CBC). However, the dietary content is not very well characterized. A typical week menu for the Altiplano workers is described, and vague references are made to an un-published pilot assessment of the diets of local workers, finding that these consume less protein and energy. Perhaps this is all that could be said at present, but some more quantitative information on the performed pilot investigation would strengthen the argumentation and the manuscript.

I am not a nutritionist, and think it could be valuable to have more information on the nutrients available and accessible in the diet. The supplemental table with a menu indicate Altiplano workers have good access to protein, iron, vitamin B12, vitamin A and folic acid in their diet, as they are served meat or eggs at least once daily. At least this is my interpretation of this menu from more or less a lay-man perspective. A nutritionist could perhaps analyze the micronutrient content of this kind of diet and add an interpretation on this to the present manuscript.

4.4. Limitations. Page 11, Line 410-411, As the authors point out, longitudinal information on nutritional indicators would be interesting in order to understand the (in)adequacy of calorie availability for the workers during harvest, and the absence of this is in the present study is a limitation. The authors refer to a previous study they have conducted on cross-harvest changes in body composition (ref 26), but this citation is dysfunctional. In the reference list there is just an author name and a title, no journal, conference name or any such information. Using Google, one cannot identify a study with that name online. I think it might have been an abstract at the Third International Workshop on Chronic Kidney Diseases Uncertain/Non-traditional Etiology in Mesoamerica and Other Regions in San José, Costa Rica, 2019, but all readers cannot be expected to have been there. If what was presented in that abstract is not available for the general public but only those having visited that conference, I think reporting on those data more completely in the present manuscript would be needed. The longitudinal information provided in that analysis would substantially improve the present manuscript.

Also, I see no reason why the authors do not include the longitudinal information collected on eGFR and weight at baseline, to look at change in these parameters in relation to the other outcomes. It would strengthen the understanding of links between these parameters and anemia.

Line 437: The authors have previously reported (ref 34, table 2) that 80% (!) of workers have low serum osmolality (<280 mmol/kg) after a workday also when drinking 10 L of rehydration solution, meaning hyperhydration was very common among these workers at that point in time, something which would have led to hemodilution and thereby falsely low hematocrit/Hb. They now believe it is less of a problem as they believe the amount of plain water consumed by workers is less than at that point in time (there is 1 years between these studies), but they have not evaluated whether this is actually the case. I think they should acknowledge how widespread hyperhydration was in the study population just one year previously by actually stating the proportion with low serum osmolality at the end of the work-shift then, and better describe why they think this has changed dramatically in the one year between these studies. In ref 34, workers drank on average between 15 L (when drinking 2.5 L of electrolyte solution) and 9 L (when drinking 10 L of electrolyte solution) of plain water per shift. How much plain water do they drink now per shift? What type of efforts have Pantaleon made on this point? As mentioned below, the hydration practices needs to be better described: how much electrolyte solution are workers instructed to drink per hour, are they instructed to drink plain water too, is it available ad libitum, or is it discouraged? If hydration practices in some way is incentivized, as it has been previously at Pantaleon through lottery tickets for those with diluted urine, this needs to be described.

Line 442 - on smoking as a cause of lower hemoglobin concentrations. It is well known that the opposite is true, hemoglobin levels are typically higher in smokers. That is also the finding in the cited article by Nordenberg.

One concern I have is whether the continuous consumption of rapidly absorbable sugars drunk by the workers as rehydration solution could possibly increase HbA1c levels, even in the absence of insulin resistance, simply because so much sugar continuously enter the circulation. It would be interesting to hear the authors view on this, and whether this can be ruled out as an explanation to the relatively high HbA1c seen here. There is not much mentioning of rehydration solution intake in the present manuscript. In previous reports by the authors from Guatemala, huge intakes of rehydration solution have been reported (~10 L daily), and this, and the content of this solution, is important to mention in relation to this analysis of nutritional aspects of the sugarcane workers’ situation. If workers continuously are drinking sugary drinks, is not HbA1C likely to increase, even in the absence of prediabetes/insulin resistance? At least among diabetics, drinking sugary drinks leads to a massive rise in HbA1c, and I wonder if that to some extent will not happen in non-diabetics too, especially if the intake is large and prolonged as in this population. If they are continuously in a prandial/post-prandial state for several hours each day and blood monosaccharide levels therefore are elevated for prolonged period of times, a large proportion of HbA is likely to become glycated, i.e. HbA1c.

Another concern which deserves mentioning as a limitation is the use of creatinine-based eGFR at the end of a workday. The relationship between this value and actual reduced kidney function/injury to me is questionable, as serum creatinine would be expected to increase from muscle work. Further, the sampling situation is not the steady-state situation which it ideally should be when GFR is estimated from creatinine. The authors have previously reported how large the difference is between eGFR assessed using serum creatinine at pre- and post-shift https://www.ncbi.nlm.nih.gov/pmc/articles/PMC6416034/figure/F3/ (i.e. in median ~20 ml/min/1.73m2).  “Current eGFR equations estimate kidney function when the plasma creatinine is stable, but do not work if the plasma creatinine is changing rapidly” as it says in an interesting paper on calculation of eGFR when creatinine is changing rapidly (https://jasn.asnjournals.org/content/jnephrol/24/6/877.full.pdf). I recognize that there is no possibility for the authors to change this now, but the problem deserves being mentioned. If the authors decide to utilize the pre-harvest eGFR measurements, then this difference between pre- and post-shift eGFR needs even more consideration. However, already now (lines 323-325) the authors compare pre-shift pre-harvest samples to post-shift mid-harvest samples and the difference between these sampling situations needs to be considered in light of what the authors have previously reported on cross-shift changes.

The comparison with proportion eGFR <90 in their previous studies (ref 14), line 327-328 are to pre-shift values. Considering that 12% in this study is at post-shift sampling and a 20 ml/min/1.73m2 decrease across the work shift, I am not sure there is a difference really.

Ethics

It is only stated that an ethical review was conducted in Colorado, US, while the research was conducted in Guatemala. If there is no obvious reason why it could not be, ethical review should have been conducted also in Guatemala as that is where the great majority of research activities were conducted. The only thing which happened in Colorado was analysis of de-identified data, the ethically least questionable aspect of this research. The authors need to explain why ethical review was not conducted in Guatemala.

Further, the study was designated as non-human subjects research. If I do not completely misunderstand the study design, there were definitely human subjects involved, and the need for informed consent seem obvious. The reason that the research was considered as non-human subject research and the need for informed consent was waived needs to be explained.

Author Response

Thank you very much for your comments. Please see the attachment for our response. 

Sincerely,

Manuscript co-authors

Reviewer 2 Report

This is a great paper that adds to our understanding of CKDu and the health status of workers at risk for CKDu. It provides insight to another potential contributor to CKDu.

Author Response

Thank you very much for the review of our manuscript. We appreciate your comments. 

Sincerely,

Manuscript co-authors